

# Variations in the Biological Pump through the Miocene: Evidence from organic carbon burial in Pacific Ocean sediments.

Mitchell Lyle

Annette Olivarez Lyle

College of Earth, Ocean, and Atmospheric Sciences, Oregon State University, CEOAS Admin Bldg 104, Corvallis Oregon 97333, USA

*Correspondence to*: Mitchell Lyle (Lylem@oregonstate.edu)

**Abstract.** The biological pump, defined as the marine biological production and sedimentation of particulate organic carbon ($C_{org}$), is a fundamental process to fix atmospheric carbon dioxide in the oceans, transfer carbon away from the atmosphere to the deep ocean, and maintain the $CO_2$ level of the atmosphere. The level of carbon sequestration by the biological pump has varied throughout the last 50 million years, from particularly weak in the warm Eocene to much stronger in the Holocene. However,

persistently warm climates in the more recent past, e.g., the Miocene Climate Optimum (MCO; 17 million years ago [Ma] to 13.8 Ma) also have affected the biological sequestration of carbon. A series of scientific ocean drill sites from the equatorial Pacific contain very low sedimentary $C_{org}$ % in the period prior to 14 Ma but higher and much more variable $C_{org}$ % afterward. Although lower absolute productivity may have contributed to the lower $C_{org}$ burial at the MCO, higher relative $C_{org}$ degradation also occurred. Ratios of $C_{org}$ to other productivity indicators indicate higher relative loss of $C_{org}$. Temperature records imply that

the higher $C_{org}$ degradation occurred in the upper water column, and global cooling strengthened the biological pump but led to more variability in burial. Similar records of low $C_{org}$ at the MCO can be found in the North Pacific, which suggest this was a global—rather than regional—change. A weakened biological pump during warm climate intervals helps to sustain periods of global warmth.

**1 Introduction**

The sequestration of organic carbon in pelagic sediments sums all the biological, physical, and chemical processes; from carbon fixation by photosynthesis, water-column transport and degradation, to ultimate sea floor deposition and burial. Heterotrophic consumption as well as inorganic oxidation greatly reduces the total particulate organic carbon ($C_{org}$) mass falling from the surface and its reactivity. $C_{org}$ that is eventually buried in pelagic sediments is protected from further attack because the

remainder is relatively recalcitrant after all the water column degradation and because $C_{org}$ binds to sediments and is less accessible for further degradation (Hedges and Keil, 1995; Mayer, 1995). Buried organic matter represents a small fraction, less than 1%, of the original primary productivity in the pelagic realm (Suess, 1980, Muller and Suess, 1979; Martin et al., 1987). In the Pleistocene pelagic equatorial Pacific, $C_{org}$ content found in surface sediments is typically low, ~ 0.2% (Murray and Leinen, 1993) while ocean margin surface sediments contain much higher $C_{org}$ (between 1-2%). The difference results from lower

primary productivity in the equatorial Pacific, deep waters (>4 km for the equatorial Pacific vs <1 km for the ocean margins), and much slower sedimentation rates that allow the $C_{org}$ to be exposed to oxygenated sea water for thousands of years.

There is evidence that during eras of global warmth $C_{org}$ degradation tends to be larger and occurs higher up in the water column (John et al, 2014; Boscolo-Golazzo et al, 2021). Upper water column $C_{org}$ degradation results in poorer sequestration of





atmospheric $CO_2$, higher atmospheric $CO_2$ in dynamic equilibrium with the oceans, and higher nutrient levels in upwelled waters. Unlike the last century, where there is intense global warming in the surface above a cold ocean, there is not necessarily a strong oxygen minimum during stable warm periods. Long periods of warming deepen the permanent thermocline and weaken the pycnocline, allowing more extensive mixing of oxygen into the ocean in contrast to the modern condition, where surface warming strengthens the temperature and density contrast between the mixed layer and colder deeper waters. The resulting

strong pycnocline in the modern ocean restricts mixing and oxygen transport downward. Higher dissolved oxygen and higher water temperatures found in long warm intervals thus intensifies $C_{org}$ degradation in surface waters and reduces burial rates. Nutrients and $CO_2$ resulting from $C_{org}$ degradation cycle back to the surface faster under warm earth conditions. Surface water that is subducted to shallow depths often reappears at the surface in decades rather than centuries for waters that are cycled to abyssal depths. Therefore, the biological pump weakens under longer intervals of global warming (Boyd, 2015). Indeed, a recent

paper by Li et al (2023) estimated global $C_{org}$ burial in the Neogene and found much less $C_{org}$ burial than expected during the Miocene Climate Optimum (MCO) interval.

The hypothesis that $C_{org}$ degradation occurred much shallower in the water column during warm ocean conditions is supported by John et al. (2014). They measured ocean depth gradients in carbon isotopes measured on planktonic foraminifera compared to

earth system model representations for the Paleocene-Eocene Thermal Maximum (PETM) and during the greenhouse Eocene to conclude that the Eocene had shallower $C_{org}$ degradation and recycling. Similarly, Boscolo-Galazzo et al (2021) examined the change in the abundance of planktonic foraminifera living at different depths in the ocean in the period from 15 to 0 Ma and used stable carbon isotope values of the foraminiferal tests, combined with earth system models, to monitor changes in carbon flux through the water column. When combined with modeling of temperature dependent $C_{org}$ degradation, their results were

consistent with cooling having caused more particulate $C_{org}$ rain to penetrate deeper into the water column to affect a stronger biological pump.

The biological pump can also be studied by examining the $C_{org}$ buried in the sediments. The $C_{org}$ mass accumulation rate (MAR; burial flux) depends upon all the processes in the water column above the sediments and is the obverse of measurements that could be made in the water column because $C_{org}$ does not sprout from the sediments below. It is a measure of the integrated

effectiveness of the biological pump. While high primary productivity increases the flow of particulate $C_{org}$ to the sea floor, increases in heterotrophic consumption by zooplankton and microbes, reduce the resulting downward flux. Because of the fundamental temperature dependence of metabolic activity (e.g., see Brown et al, 2004), higher water temperatures increases the rate of $C_{org}$ degradation by zooplankton and microbes, and decreases $C_{org}$ burial (e.g., Boscolo-Galazo et al., 2018).


Preservation of $C_{org}$ in sediments can also be studied by comparing it to other paleoproductivity measures that are better preserved. For example, Biogenic barium (bio-Ba) deposition is strongly correlated to $C_{org}$ rain from the euphotic zone (Dymond and Collier, 1996) and is better and more consistently preserved at the sea floor under typical oxygenated conditions (Dehairs et al, 1980; Dymond et al., 1992; Ganeshram et al., 2003). Lyle and Baldauf (2015) studied relative $CaCO_3$ dissolution in this way,

using ratios of $CaCO_3$ to Ba. Under warm earth conditions associated with early Cenozoic greenhouse conditions (Olivarez Lyle and Lyle, 2005, 2006, Lyle et al., 2005) buried $C_{org}$ % was extremely low (Eocene average of 0.03%). Olivarez Lyle and Lyle (2005, 2006) used bio-Ba burial to estimate an 'expected' $C_{org}$ MAR assuming modern conditions. All $C_{org}$ burial during the Eocene was much lower than modern $C_{org}$ burial relative to production, indicating much higher heterotrophic consumption of $C_{org}$ within the water column. The ratio of $C_{org}$ to Ba is thus an indicator of relative $C_{org}$ preservation.


We produced long-term, low-resolution $C_{org}$ and $CaCO_3$ data between 2008 and 2013 for drill sites extending into the Miocene in the Pacific Ocean (Fig 1, Table I). In this paper we report discrete $C_{org}$ measurements we have made on 3 scientific drill sites from the eastern and central equatorial Pacific (ODP Site 574, and IODP Sites U1337, and U1338) and combine them with data from the Ontong Java Plateau in the western equatorial Pacific (ODP Sites 806 and 807; Stax and Stein, 1993). We also report

briefly on two sites from the northwest Pacific (Sites 884 and 1208) to show that changes in $C_{org}$ burial were not limited to the equatorial region. We use these data to study changes in the biological carbon pump from the MCO to the present by comparing the patterns of $C_{org}$ preservation through time and, where we have data, by comparing how $C_{org}$ survives relative to bio-Ba, a better preserved paleoproductivity indicator.

| Site | location | N is +<br>Latitude | E is +<br>Longitude | water<br>depth (m) | 5 Ma<br>Latitude | Longitude | 10 Ma<br>Latitude | Longitude | 15 Ma<br>Latitude | Longitude | 20 Ma<br>Latitude | Longitude | Notes |
|------|----------|---------|-----------|-----------|---------|-----------|---------|-----------|---------|-----------|---------|-----------|-------|
| U1338 | E. equatorial Pacific | 2.508 | -117.969 | 4205 | 1.43 | -113.79 | 0.49 | -110.00 | -0.37 | -106.31 | ... | ... | eq crossing at 13 Ma |
| U1337 | E. equatorial Pacific | 3.833 | -123.206 | 4466 | 2.55 | -119.07 | 1.50 | -115.31 | 0.53 | -111.63 | -0.36 | -107.93 | eq crossing at 18 Ma |
| 574 | E. equatorial Pacific | 4.209 | -133.33 | 4571 | 2.60 | -129.18 | 1.35 | -125.41 | 0.21 | -121.73 | -0.86 | -118.02 | eq crossing at 16 Ma |
| 806 | W. equatorial Pacific | 0.319 | 159.361 | 2521 | -1.99 | 163.62 | -3.41 | 167.39 | -4.63 | 171.04 | -5.90 | 174.69 | eq crossing at <1 Ma |
| 807 | W. equatorial Pacific | 3.607 | 159.625 | 2804 | 1.33 | 160.94 | -0.05 | 164.76 | -1.24 | 168.45 | -2.48 | 172.13 | eq crossing at 10 Ma |
| 1208 | N. Pacific, Shatsky Rise | 36.127 | 158.208 | 3346 | 33.83 | 163.05 | 32.42 | 167.37 | ... | ... | ... | ... | Hiatus to Paleocene >12 Ma |
| 884 | N. Pacific, Detroit Seamount | 51.45 | 168.337 | 3827 | 49.05 | 173.09 | 47.52 | 177.48 | 46.19 | -178.30 | 44.82 | -174.21 | eq crossing at 10 Ma |


**Table I: Locations of drill sites discussed in this paper and paleo-locations of the sites through the Miocene**

Li et al. (2023) used a global set of mass accumulation rates to show that there was a global low in $C_{org}$ MAR during the MCO, with a primary objective of the paper to test the "Monterey hypothesis" that high $d^{13}C$ between 17 and 13 Ma resulted from

higher $C_{org}$ burial. They found that the MCO interval had globally low $C_{org}$ burial and was not a cause for high $d^{13}C$ at that time. The objective in this paper is different. We chose to examine a focused region (the equatorial Pacific) and use additional sedimentary data to separate changes in production from changes in preservation in the ultimate $C_{org}$ record. We examine 4 hypotheses for the low $C_{org}$ burial at the MCO— (1) low primary productivity, (2) $C_{org}$ degradation in warmer deep waters, (3) $C_{org}$ degradation in warmer surface waters, and (4) a fundamental change in the proxy-Corg relationship. We find that the low

$C_{org}$ MAR in the equatorial Pacific results largely from better preservation of sedimentary $C_{org}$ as the earth cooled through lower degradation in the surface ocean.

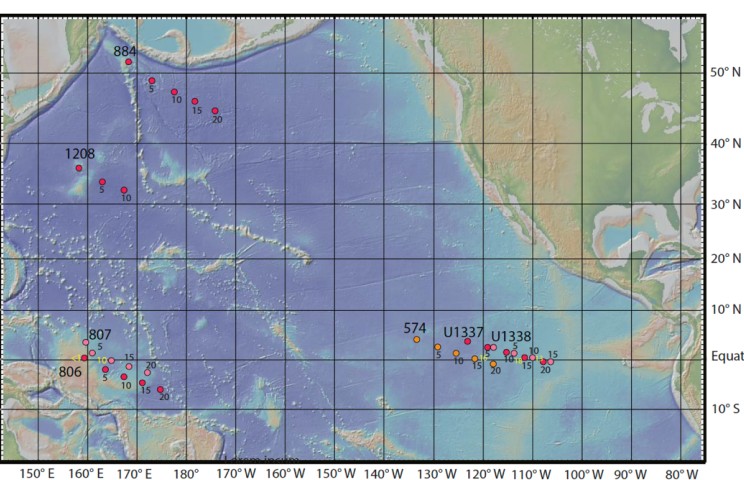



**Figure 1: Drill site locations discussed in this paper. Paleo-locations for each site are shown at 5 Myr intervals, using a fixed hotspot rotation pole for the Pacific tectonic plate. The equatorial ODP drill sites 806 and 807 were drilled on ODP leg 130 (Stax and Stein, 1993. Sites U1337 and U1338 were drilled on IODP expedition 321 (Lyle et al, 2019; Pälike et al, 2010). Site 574 was drilled on DSDP Leg 85 (Mayer et al, 1985). Two other sites are briefly discussed—Site 1208 on the Shatsky Rise (Bralower et al., 2002) and Site 884 near Detroit Seamount at the northern end of the Hawaii-Emperor chain (Rea et al., 1993). Yellow numbers indicate the age of each site's equator crossing, a local high in primary productivity and sediment deposition.**


## 2 Analytical Methods

We measured over 1,600 samples (1,215 presented here plus 388 previously published) for $C_{org}$ and $CaCO_3$, from 5 Sites of the Deep Sea Drilling Project, Ocean Drilling Program, and Integrated Ocean Drilling Program (Sites 574, 884, 1208, U1337, and U1338, see Fig. 1).  Freeze-dried sediment samples were analyzed using coulometry and furnace methodologies presented in

Lyle et al., (2000) which provide accurate $C_{org}$ values at very low levels. First, Total carbon of the sediment sample was measured through combustion in a 1000°C furnace and analyzed using a coulometer.  Organic carbon was determined by acidifying a second, larger aliquot of the sample before analysis to remove the $CaCO_3$-based $CO_2$, then analysing the remaining residue via combustion in pure oxygen at 1000°C via coulometry.  From these two measurements, the $CaCO_3$ fraction is calculated based on the difference between the $CO_2$ measured for Total carbon and the organic carbon fraction. We found that

this method is accurate for estimating $CaCO_3$ down to the <1% level and for $C_{org}$ at the 0.01% level (Olivarez Lyle and Lyle, 2005).

We typically do not use $C_{org}$ data from DSDP or ODP that were analysed shipboard because the method is inaccurate below 0.3% $C_{org}$ when in the presence of significant sedimentary $CaCO_3$ (Olivarez Lyle and Lyle, 2005). Using our method of analysis,

data for Site 574 sediments older than 12 Ma are reported in Piela et al (2012). $C_{org}$ data spanning 12 to 2 Ma were analyzed later and are published here. Similarly, $C_{org}$ data from Sites 884 and 1208 are also first published here. For Site 884, we used shipboard $C_{org}$ when the $CaCO_3$ % was less than 15% because the error of the shipboard method at this level of $CaCO_3$ is relatively small (<1.8 wt.% carbon from the $CaCO_3$ fraction vs ~0.1 to 0.4% for the $C_{org}$ fraction). Utilising these shipboard data at Site 884 helped us to complete a full record of deposition. For Sites U1337 and U1338, $C_{org}$ data are reported in Wilson (2014)

but were not discussed in her thesis. All the data are reported here in the Supplemental Material; Tables S1 to S20.

We include data from Stax and Stein (1993) for Sites 806 and 807. They measured $C_{org}$ by dissolving an unweighed split of the sample with HCl and measuring $C_{org}$ after dissolving the $CaCO_3$. They then corrected the split $C_{org}$ to that of the total sample by measuring total carbon on a separate split of the sample and calculating the relative proportion of $C_{org}$. Using this method, they

could dissolve a much larger split of sediment for $C_{org}$, so that they were accurate at low $C_{org}$ contents while also being able to measure other $C_{org}$ properties.

The net deposition and preservation of $C_{org}$ at Sites U1337, U1338, and Site 574 is estimated from ratios to the barium content in the sediments. Extensive work has shown this element is precipitated in the water column as $BaSO_4$ via organic matter oxidation,

is well-preserved, and importantly, is a good proxy for primary productivity (Dymond et al, 1992, Dymond and Collier, 1996; Ganeshram et al, 2003). Although barium sulfate can form through other processes, in equatorial Pacific sediments, the Ba content is primarily biogenic in origin and not diluted by other Ba-containing components, for example, by barium in terrigenous clays (Piela et al., 2012; Lyle and Baldauf, 2015)). We calibrated Ba measurements from X-ray fluorescence (XRF) (normalized





median-scaled 'NMS', see Lyle et al, 2012) to discrete ICP-MS analyses for Sites U1337 and U1338 (Wilson, 2014). At Site 574

we used ICP-MS barium data from Piela et al. (2012) for sediments older than 12 Ma.

We use total XRF-estimated total $BaSO_4$ rather than bio-Ba to ratio to other biogenic elements in this paper to investigate $C_{org}$ preservation. Total barium is justified, as stated, because the equatorial Pacific sediments at Sites U1337 and U1338 primarily consist of biogenic remains and contain little clays, so the majority of Ba is biogenic in origin. We further tested this assumption

based on an estimate of terrigenous, authigenic oxide, and authigenic clay-based Ba at Site U1338 and found that biogenic $BaSO_4$ averaged 93% ±4% of total $BaSO_4$. The greatest error in our biogenic barium estimates occurs when this fraction is at its lowest in the presence of relatively abundant clays, and are located primarily at the tops of the sites. Such a bias towards shallow core depths consequently skews the $C_{org}/Ba$ ratio to lower values in younger sediments where the $C_{org}/Ba$ and estimated preservation is typically highest.


### 3 Age Models and Mass Accumulation Rates (MAR)

For all sites we calculated $C_{org}$ mass accumulation rates (MAR), equivalent to $C_{org}$ burial flux, as another indication of the rate of $C_{org}$ deposition and burial. MAR has units of mass per unit area per unit time and typical reported as "grams /($cm^2$ x $10^3$yr)". For any component in a sedimentary mixture, calculating its mass accumulation rate eliminates artifacts in the resulting data profile

over depth/time that are necessarily caused by variable deposition of the other sedimentary components. Converting weight % data to MAR is particularly important for evaluating changes in minor components such as $C_{org}$ because change is deposition of the major components can dominate and distort actual changes of the component of interest. However, MAR calculations are subject to errors (for example, sedimentation rates) that are primarily caused by an imprecise age model. This issue is addressed below.


We developed age models for all sites to calculate MAR to further study rates of burial. See the Supplemental Material for details and for data from each of the sites. MARs are developed by making an age model (age vs. depth profile) that is differentiated to calculate a sedimentation rate (thickness of sediment deposited per unit time; cm per kyr) over the time interval of interest. Sedimentation rate, multiplied by the Dry Bulk Density results in a bulk MAR (g solid/$cm^2$/kyr) for the sediment

over a given age interval. Individual component MARs are calculated by multiplying the bulk MAR by the weight fraction (wt %/100) of the component in the dry sample.

### 3.1 Sites U1337, U1338

For Site U1337 (24 Ma crust) we used an astro-chronologically-tuned age model to 20 Ma by combining the Drury et al (2017,

2018) age model to 8.2 Ma and correlating the 20 Myr combined isotope record from Site U1337 (Holbourn et al., 2015; Tian et al., 2018) to the stable isotope stack from Westerhold et al (2020) to 20 Ma. For Site U1338 (18 Ma crust) we used the established age model to 8.2 Ma (Lyle et al., 2019). The U1338 ages greater than 8.2 Ma were derived by correlating the U1338 $CaCO_3$ profile to that from Site U1337, which is justified because $CaCO_3$ profiles are very similar across the central and eastern Pacific (Lyle et al, 2019; Mayer,1991).


### 3.2 Site 574

Site 574 (35 Ma crust) was drilled in 1985 during Leg 85 of the Deep Sea Drilling Project and lies about 1000 km west of Site U1337. It was one of the first sites to have hydraulically piston cored sediments. We XRF-scanned the upper 5 cores of Holes





574 and 574A and used these data and the GRA bulk density record in deeper sections to make a new splice to ~17.4 Ma (225 m composite depth, the base of Holes 574 and 574A) using the *Code for Ocean Drilling Data* software (CODD; https://www.codd-home.net/, last access: 08 August 2023). The new splice was then correlated to Site U1337 by a $CaCO_3$ profile produced by using the Site 574 GRA bulk density to estimate $CaCO_3$ content. In the eastern and central Pacific Ocean, the GRA bulk density is highly correlated with changes in $CaCO_3$ content (Mayer, 1991). The noncarbonate fraction in the equatorial Pacific is primarily low density, high porosity biogenic silica remains of diatoms. The GRA bulk density data is also much higher

resolution than the discrete carbonate analyses used for calibration. The spliced GRA density record from Site 574 is in the supplemental tables with assigned ages (Supplemental Table S2), as well as the new splice developed for Site 574 (Supplemental Table S1).

### 3.3 Sites 806 and 807

We used polynomial fits to ages updated to the biostratigraphic ages used for Sites U1337 and U1338 (Expedition 320/321 scientists, 2010) for the biostratigraphic datum levels for Sites 806 and 807 reported in the Initial Reports volume (Kroenke et al., 1991). Such fits are less accurate than a direct correlation, but these sites do not have long stable isotope records, nor was it possible to correlate the carbonate records to the eastern Pacific. While not as accurate as the other age models, ages should still be good to ± 0.2 Ma.


### 3.4 Sites 1208 and 884

Age-depth profiles for Sites 1208 (Shatsky Rise, to 12.4 Ma) and 884 (Detroit Seamount, at the north end of the Hawaii-Emperor seamount chain, to 19.7 Ma) were made using linear interpolations between magnetochrons. We used Evans (2006) for magnetochrons from Site 1208, and shipboard paleomagnetic data for Site 884 (Rea et al., 1993). All magnetochrons have

updated ages based on Westerhold et al. (2020). Accuracy of ages should be better than 0.1 Ma. The profiles are listed in the supplemental material. Site 1208 was located on the Shatsky Rise itself and had an extreme slowdown in sedimentation from the middle Miocene through the late Cretaceous which may have affected the early part of the profile.

Bulk MAR for each site was calculated as the product of sedimentation rate (from the age/depth profiles) and the dry bulk

density to yield the total mass of sediment deposited over a given age interval (bulk MAR). Dry bulk density was estimated by correlating discrete physical properties data from each drill site to the reported GRA wet bulk density data. We used the correlation to develop a higher resolution dry bulk density profile from the GRA data.

### 4 Results

### 4.1 Organic Carbon Variations:

There is a striking difference between the the $C_{org}$ records from the Pacific equatorial sites (U1338, U1337, 574, 806 and 807) older than ~13 Ma  and the younger sections, as illustrated by Figure 2. Low sedimentary $C_{org}$ %, $C_{org}$ MAR, and $C_{org}/Ba$ within the MCO changes to higher and more scattered values between 14 and 12 Ma, after the MCO. $C_{org}$ wt.% was much lower in the older record as compared to the Pliocene and Pleistocene (Fig 2A). Within the MCO, between 17 and 14 Ma, $C_{org}$ wt% in these 5

sites averaged 0.043 ± 0.014 %, in contrast to a nearly 3-fold increase (0.124 ± 0.058 %) in the Pleistocene and Pliocene  (0-4 Ma) and we note the MCO values are more similar to the average  measured for Eocene equatorial Pacific  ($C_{org}$ roughly 0.03 wt%, >34 Ma; Olivarez Lyle and Lyle, 2006). Both the concentration of sedimentary $C_{org}$ and the variability were much lower in the period prior to 14 Ma.





High dilution by other sedimentary components can lower $C_{org}$%, but dilution in the equatorial Pacific requires higher preservation and burial of either biogenic $CaCO_3$ or biogenic $SiO_2$ relative to modern conditions. Nonetheless, this possibility can be evaluated by calculating the $C_{org}$ MAR which removes the dilution effect and is a measure of the actual burial flux. Fig 2B shows the $C_{org}$ MAR data for the 5 equatorial Pacific sites. We note a relatively low and consistent $C_{org}$ MAR between 18 and 14 Ma. Somewhat higher levels of $C_{org}$ MAR occur prior to 19 Ma and higher but very scattered MAR after 14 Ma.


**4.2 Variations in Biogenic Barium**

Ba was measured at 3 of the sites, all in the high-productivity region of the eastern Pacific. The $C_{org}$/Ba ratio compares the burial of the more labile $C_{org}$ (<1% $C_{org}$ preserved) to that of the better-preserved biogenic Ba (~30% preservation, Dymond et al, 1992, Dymond and Lyle, 1994). Since Ba has a linear relationship with $C_{org}$ in the particulate rain of the modern equatorial Pacific

Ocean (Dymond and Collier, 1996), changes in the $C_{org}$/Ba ratio is interpreted to reflect the relative preservation of $C_{org}$. Fig 2C shows that there was lower $C_{org}$ burial relative to Ba prior to 14 Ma than later in the records.

Ba was measured by ICPMS at Site 574 for the sediment column older than 12 Ma (Piela et al, 2012). Sites U1337 and U1338 have Ba measured by scanning XRF for the entire composite sediment column (Lyle et al., 2012; Wilson, 2014, Lyle and

Baldauf, 2015). All have low $C_{org}/Ba$ ratios (poor $C_{org}$ preservation) prior to 14 Ma in the MCO, and higher ratios with much greater variation since then.

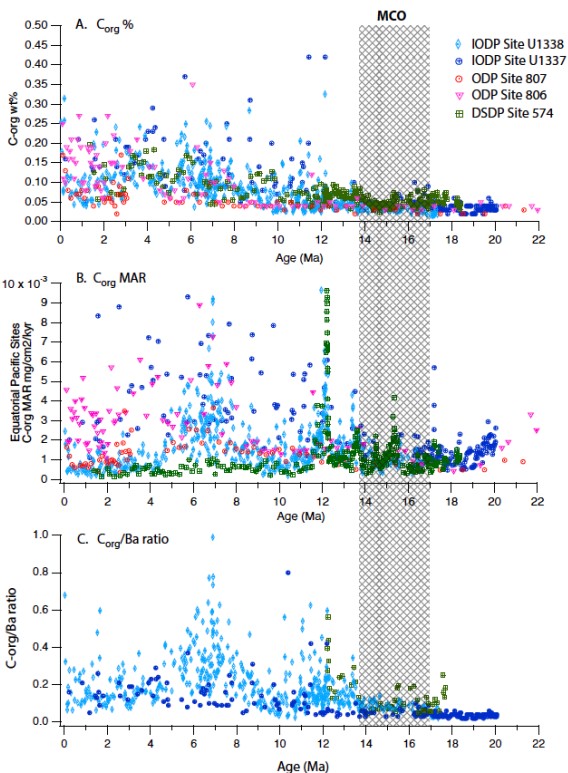

**Figure 2: Corg % (A) Corg MAR (B) and $C_{org}$/Ba (C) for the equatorial sites 574, 806, 807, U1337, and U338 to show evidence of low $C_{org}$ contents and poor $C_{org}$ preservation over the Miocene Climate Optimum (MCO, 17-13.8 Ma).**




Low $C_{org}$ % during the MCO is not confined to the equatorial Pacific but also found in the north Pacific. We also analyzed two Sites from the North Pacific to produce organic carbon records which show increased $C_{org}$ burial as the Miocene progressed. At Sites 1208 on the Shatsky Rise, and Site 884 on Detroit Seamount (Fig 1), $C_{org}$ MAR has increased strongly post MCO (Supplemental tables S8 and S10) A straightforward interpretation of these records, however, is complicated by the fact that

there is a major increase in sedimentation rate at Site 1208 after a Paleocene hiatus which also causes increased $C_{org}$ MAR post 7.5 Ma. As such, it is unclear to what extent the local sedimentation regime played a role in the increase of $C_{org}$ MAR relative to regional levels of preservation. At Site 884, sedimentation rates increase in the early Miocene, leading to higher $C_{org}$ MAR after the MCO (see supplemental material Fig S5). There is no reason to suspect that the Site 884 record was strongly influenced by changes in the local sedimentary environment, however, there is overall much higher clay deposition here that might have

affected $C_{org}$ burial. In summary, these two Sites suggest a general increase in $C_{org}$ MAR throughout the Pacific after the MCO, but more records need to be produced and evaluated to support the hypothesis. However, it is important to highlight the data from a recent study by Li et al (2023) who found surprisingly low $C_{org}$ burial across the MCO worldwide, in support of the hypothesis that a global slowdown in $C_{org}$ deposition occurred during the MCO.

**5 Discussion and Implications**

We expect variability in the deposition of biogenic particulate matter resulting from large scale tectonic-biogeographic processes and from global intervals of high productivity, as well as from regional variation in primary productivity. We note that $C_{org}$ depositional variability from 12 to 0 Ma is high, between 0.03 and 0.40 wt.%. The variation is partly caused by the geographic position of the site relative to the high equatorial productivity zone, and partly from the presence of high productivity intervals

since the MCO. Some may also result from protection of $C_{org}$ by the deposition of other sedimentary components. Nevertheless, we note that more slowly accumulating sediments of the Pliocene and Pleistocene tend to have high $C_{org}$ % (Figs 3 and 4) but without corresponding high biogenic MARs. In other words, factors other than high productivity have caused better preservation of $C_{org}$ between 8 and 0 Ma since high $C_{org}$ % is not necessarily found with other indicators of high paleoproductivity. All records in the equatorial Pacific show evidence for low $C_{org}$ % and $C_{org}$ MAR during the MCO (Fig 2), while sediments from the

Pliocene and Holocene have relatively high $C_{org}$ % even though many of the sites exhibit low $C_{org}$ MAR.

We interpret the scatter to reflect the localized response to the timing of productivity drivers combined with better $C_{org}$ preservation in more recent times. There are 2 major productivity drivers in the equatorial Pacific: First, when the site is carried across the equatorial high productivity region by tectonic motion of the Pacific plate (line of high productivity for the Neogene;

Lyle, 2003, Moore et al., 2004), and second, during high productivity intervals which typically are limited in both space and time. One example is the late Miocene Biogenic Bloom, between 8 and 4.5 Ma (Dickens and Owen; 1999; Diester-Haas et al, 2002; Lyle and Baldauf, 2015; Karatsolis et al., 2022; Gastaldello et al., 2023), and an earlier high productivity interval between 13 and 10.5 Ma (Lyle and Baldauf, 2015) and extending to ~14 Ma (Holbourn et al., 2014).

**5.1 Tectonic passage through the equatorial productivity zone and equatorial primary productivity**

Modern studies of equatorial biological productivity, including direct measurements in the water column, MAR in surface sediments, and those based on interpretations of satellite color find that the particulate flux both at the surface and to sediments is highest at the equator and strongly decreases to the north and south (Wyrtki, 1981; Chavez and Barber, 1987; Dugdale et al., 1992; Murray and Leinen, 1993; Honjo et al., 1995; Behrenfeld et al, 2005). This pattern of high equatorial biogenic flux has





been found throughout the Neogene (Moore et al., 2004, Berger, 1973). The records are strong evidence that the equatorial divergence driven by the SE trade winds crossing the equator has been a persistent feature of the Cenozoic oceans that causes high primary productivity at the equator. The magnitude of equatorial productivity has not remained constant through time, however, but has waxed and waned along with global climate change. We expect to find a change to higher productivity and deposition of biogenic sediments as the movement of the Pacific Plate brings a drill site into position at the equator, and a

decrease in biogenic MARs as the site is moved away from the equator.

For example, Piela et al (2012) found that high biogenic silica mass accumulation rate, bio-Si MAR, and barium mass accumulation rate, Ba MAR, occurred during the Site 574 paleo-equator crossing at 16.25 Ma, despite low $C_{org}$ MAR and low $CaCO_3$ MAR. They hypothesized that the dissolution of $CaCO_3$ exposed more $C_{org}$ in surface sediments to potential degradation

and reduced $C_{org}$ with respect to other productivity signals. At site U1338, high burial rates of biogenic components other than $C_{org}$ coincide with the period that Site U1338 was within ± 0.5° of the paleo-equator during the middle Miocene (= 55 km distance, 16-10 Ma, Fig 3). $C_{org}$ MAR, unlike the other biogenic components, was not as enhanced during the MCO, implying that $C_{org}$ burial was minimized during the MCO and immediately after. Also, at Site U1338, the high $C_{org}$ MAR and opal MAR at 12 Ma are roughly equivalent to modern MARs in surface sediments as reported by Murray and Leinen (1993). $CaCO_3$ MAR is

not only affected by high productivity but also by changes in carbonate dissolution through time. Nevertheless, much of the variation in $CaCO_3$ at Site U1338 is common with variations in both opal and $BaSO_4$ MARs, indicating a strong productivity signal at the site.

The site U1337 oxygen isotope record is the 15-20 Ma part of the Westerhold et al. (2020) Cenozoic oxygen isotope splice and

thus, along with Site U1338, represents the MCO interval in the equatorial Pacific. The sediments from Site U1337 (Fig 4) crossed the equator from the south-east to north-west at an earlier time than Site U1338 (from 21 to 15 Ma versus 16 to 10 Ma, respectively) and had relatively low biogenic MARs in the older sediments as compared to later in the U1337 section. This is in part caused by sediment focusing in the younger part of the record, especially in the intervals 6.2-5.4 Ma and 4.5-3 Ma (Lyle et al., 2019, and its supplemental material). The sediment focusing is shown by anomalously high sedimentation rates for sediments

at this latitude in the Pleistocene and by erosional channels to the northeast of the site. Nevertheless, sediment focusing has not affected the time interval spanning the MCO for which we find very low $C_{org}$ % and $C_{org}$ MAR. Such low values contrast with the expected pattern of higher biogenic sedimentation beneath the high productivity region of the paleo-equator. We also note that Bio-Barium MARs at Site 1337 are 3 times higher than at Site 1338 during the MCO, implying greater primary productivity and organic carbon production, yet this signal was not preserved in the sediments.


Sites 806 and 807, in the western Pacific on Ontong Java Plateau, crossed the equatorial region at <1 Ma and 10 Ma, respectively. In the modern ocean, the upwelling signal in the far western equatorial Pacific is not nearly as strong as in the eastern equatorial Pacific (Behrenfeld et al., 2005; Rousseaux and Gregg, 2017). Site 806 has its highest $C_{org}$ MAR in the Pleistocene as expected by its recent equator crossing. However, Site 807 shows little sign of its equator crossing in $C_{org}$ MAR at

10 Ma, although it and Site 806 have a high $C_{org}$ MAR resulting from the Late Miocene Biogenic Bloom. However, both sites have low $C_{org}$ MAR during the warm MCO.





**Figure 3: A) C$_{org}$ wt %, B) C$_{org}$ MAR, C) BaSO$_4$ MAR, D) Biogenic SiO$_2$ MAR, and E) CaCO$_3$ MAR time series for Site U1338, on
ocean crust formed at 18 Ma.. High biogenic MARs are characteristic of later high productivity intervals. However, the MCO has low
C$_{org}$ contents and C$_{org}$ MAR, despite Site U1338 being relatively near the equator at that time. Paleo-equator line marks the time when
plate tectonic movement aligned the site with the equator.**

**5.2 Past high productivity intervals**

Figure 3 has time series of all biogenic MAR components at Site U1338. Note the interval from roughly 14.5 Ma to 11.5 Ma





indicates high primary production despite having low $C_{org}$ burial within the MCO. High productivity during this interval at Site U1338 was first noted by Holbourn et al. (2014), for the period near 14 Ma, suggested that enhanced equatorial Pacific biogenic silica production was driven by changes in insolation which helped to draw down high atmospheric $CO_2$ and cause cooling at the




**Figure 4: A) $C_{org}$ wt %, B) $C_{org}$ MAR, C) $BaSO_4$ MAR, D) Biogenic $SiO_2$ MAR, and E) $CaCO_3$ MAR time series for Site U1337, to the west of Site U1338, on 24 Myr ocean crust. The biogenic MAR time series are more complex here because of sediment focusing in the younger part of the record. However, the MCO has low $C_{org}$ contents and $C_{org}$ MAR, despite Site U1337 being relatively near the equator at that time. Paleo-equator marks the time when plate tectonic movement aligned the site with the equator.**






the end of the MCO. In addition, there is a well-documented global productivity interval between 8 and 4.5 Ma found globally as the "Late Miocene Biogenic Bloom", (LMBB) (Dickens and Owen, 1999; Diester-Haas et al, 2002; Lyle and Baldauf, 2015; Drury et al, 2017; Lyle et al, 2019; Karatsolis et al, 2022.) Both episodes are clearly observed at Site U1338. At Site U1337, there is similar levels of biogenic Si MAR to Site U1338 during the older interval but there is another interval filled with

laminated diatom mats between 10 and 12 Ma (200-250 m CCSF) that might represent accumulation near the subduction boundary between the South Equatorial Current and North equatorial countercurrent (Exp 320/321 Scientific Party, 2010b). So, at Site U1337, the two separate higher productivity intervals are joined by a third interval of higher deposition. Both Sites 806 and 807 show elevated $C_{org}$ MAR associated with the LMBB indicating that these global high productivity intervals affect all sites in the equatorial Pacific.


### 5.3 Four Hypotheses to explain low Corg burial at the MCO

We propose four working hypotheses for the cause of low levels of $C_{org}$ MAR along the Pacific equator during the MCO: (1) generally low primary productivity during the MCO, (2) warmer deep waters and increased $C_{org}$ degradation in the lower water column or sediment surface, (3) increased degradation of $C_{org}$ in surface waters; and (4) low apparent $C_{org}$ as an artifact of

changing proxy relationships, e.g., Ba fixation in relation to $C_{org}$ fixation in particles. In only one hypothesis (1, low primary productivity) does the relationship between $C_{org}$ production, production of productivity proxies and their ultimate burial remain the same; the others assume a change in these relationships occurred during the MCO.

### 5.3.1 Hypothesis (1): Low productivity is the primary factor for low $C_{org}$ during the MCO

Under hypothesis (1), the observed low $C_{org}$ % during the MCO reflects low average productivity during the MCO interval. However, the data do not support this hypothesis. Each of the sites were in a different position relative to the equatorial high productivity zone at the time of the MCO. Those near the equator had as low a $C_{org}$ % and $C_{org}$ MAR as those farther away. Furthermore, there is no evidence of global low productivity during the MCO. Site U1338 which was in the equatorial zone late in the MCO, has increases in $C_{org}$ % and $C_{org}$ MAR at the end of the MCO. Since other paleoproductivity indicators tended to be

high during the MCO it is apparent that low $C_{org}$ MAR is not a result from low MCO primary productivity.

In summary, while we find some evidence for somewhat lower productivity during the MCO, high primary productivity associated with drill site equator crossings are reflected in increased burial of other biogenic sediment components, but not for $C_{org}$. Therefore, we conclude that additional factors caused the lower $C_{org}$ MAR during the MCO.


### 5.3.2 Hypothesis (2): warmer deep waters and more $C_{org}$ degradation in the lower water column or sediment surface led to low $C_{org}$ MAR during the MCO

Particulate matter falls through the oceans at a rate of ~100 m/day (Honjo et al, 1982) meaning that it spends about 10 days above 1000 m in the upper water column and about a month in the deep waters below before reaching bottom, assuming an

average ocean 4 km water depth. In addition, particulate $C_{org}$ spends decades to centuries at the sediment surface before final burial. Provided that particulate $C_{org}$ has survived the passage through the upper water column, it is possible that temperatures or other processes in the lower water column have controlled burial of $C_{org}$ by temperature dependent degradation of $C_{org}$. If the change in $C_{org}$ degradation occurs primarily in the lower water column, one consequence is that the biological pump would still function, albeit at a somewhat lower rate.






There are two problems with the hypothesis for lower water column control of $C_{org}$ burial—first, most $C_{org}$ is remineralized in the upper water column in the modern oceans, so only the more recalcitrant $C_{org}$ fractions or those protected in some way survive to abyssal depths. Second, the cooling of the abyssal Pacific occurs much later than the end of the MCO (Fig 5). Any temperature-linked degradation of $C_{org}$ in deep waters should decrease only after the deep waters cool. As Fig 5 illustrates, Pacific deep waters

were only slightly warmer within the MCO than immediately after that period and stayed at a relatively constant temperature of 7° C (Lear et al, 2015) until about 6 Ma. After 6 Ma deep waters cooled gradually to a modern temperature of about 2°Ç. Therefore, we expect a $C_{org}$ signal caused primarily by deep water cooling to have a much different signal than that observed.

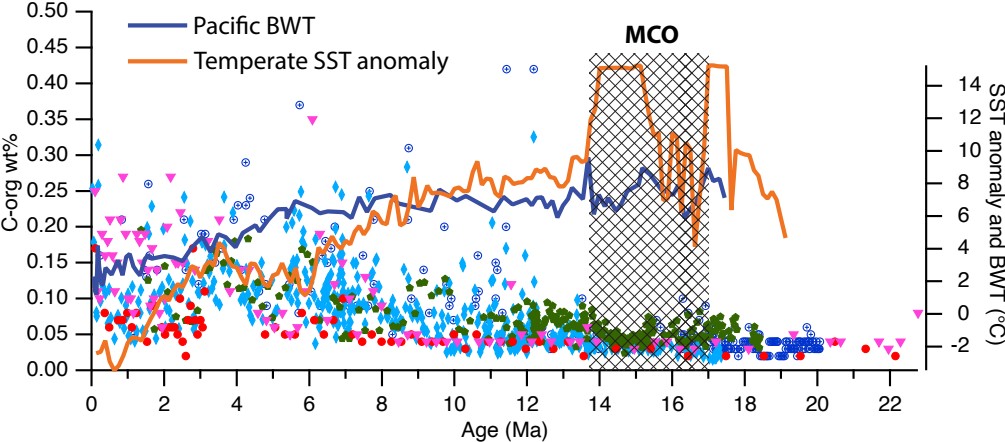

**Figure 5: $C_{org}$ wt % time series in the equatorial Pacific along with bottom water temperature (BWT, Lear et al, 2015, from Site 806) and a stacked temperate water sea surface temperature (SST, Herbert et al., 2022) expressed as anomaly from modern. We show the temperate SST anomaly record because the data are based on alkenone paleotemperatures, and older equatorial SST is higher than temperatures that can be measured by alkenones. BWT in the Pacific did not change much until after 6 Ma, too late to affect $C_{org}$ degradation in the MCO.**


The abyssal temperature change during the MCO is relatively small. Lear et al. (2015) data show that the MCO was 0.4°C warmer than the next 3 Myr of the temperature record at Site 806 (Fig 5). Kochann et al. (2017), in contrast, describe an abyssal temperature change of 1.7° at the end of the MCO at Site U1338 and 2.2°C at Site 1146 in the South China Sea. However, they did not observe consistently warm temperatures through the intervals in the MCO that they measured, in contrast to the

consistently low $C_{org}$ contents. Kochann et al. (2017) noted a transient 2.6°C warming of bottom water temperatures at the onset of peak MCO warmth (15.5 Ma) at sites U1337 and U1338. However, both Lear et al (2015) and Kochann et al. (2017) found intermittent warming during the MCO, thus inconsistent with abyssal warmth being a driver of the low $C_{org}$ over the entire MCO.

A low percentage of particulate matter derived from total productivity reaches the sea floor, and sediment trap studies have

shown that the particulate flux that reaches the sea floor is relatively rich in $C_{org}$. The deepest sediment traps in the Joint Global Ocean Flux Study (JGOFS) tropical Pacific Experiment caught a particulate flux that contains around 5 wt % $C_{org}$ (Honjo et al., 1995; particulate rain from sediment traps between 2191 m and 3618 m depth within ± 5° of the equator), which, subsequently, must degrade down to the typical content found in surface equatorial sediments, around 0.2 to 0.3% (Murray and Leinen, 1993; Prahl et al, 1989). This order of magnitude loss of organic carbon, from the deepest waters to surface sediment, appears to be a





consistent level of degradation in the pelagic equatorial Pacific, as is also shown by core top values of $C_{org}$ in the drill site data.
Interestingly, highest $C_{org}$ found in surface sediments measured in the JGOFS transect by Murray and Leinen (1993) are in off-equatorial sites, where clay is much more abundant and sedimentation rates are much lower. The lower equatorial $C_{org}$ is partly a dilution effect by additional $CaCO_3$ compensated by higher sedimentation rates near the equator (Murray and Leinen, 1993).

If high abyssal temperature were a major factor in the $C_{org}$ degradation, one might expect a major change in temperature associated with the MCO. Instead, there is only a minor abyssal temperature change at the end of the MCO (Fig 5; Lear et al, 2015). The major temperature decline occurred between 6 and 3 Ma associated with the end of the Miocene and Pliocene. Between 14 and 6 Ma, the BWT stayed between 6 and 7 °C, relatively constant, before dropping to 3° at about 2.5 Ma.

**5.3.3: Hypothesis (3): warmer surface waters and more $C_{org}$ degradation in the upper water column during the MCO**

Our preferred hypothesis is that higher degradation in surface waters during the MCO reduced $C_{org}$ burial via diminished rates of *$C_{org}$* transferred to deep waters, while post-MCO cooling led to increased opportunities for greater $C_{org}$ transfer to the abyss, partly dependent on the productivity regime over each drill site (Fig 5, Herbert et al, 2022). Note, the Herbert et al (2022) data is expressed as an anomaly, relative to modern SST, so that multiple sites that remained below alkenone saturation could be
combined into one profile.

Fig 5 illustrates the large drop, roughly 9°C, in Northern Hemisphere mid-latitude SST immediately following the MCO. Because the SST data from Herbert et al. (2022) are based on alkenones, and the equatorial SSTs prior to 12 Ma were near or above the maximum temperature that alkenones record (~29°C, Rouselle et al., 2013), the midlatitude Northern Hemisphere SST anomaly change is displayed in Figure 5. The tropical temperature change is probably smaller than that of the NH midlatitudes
and may be confined more to the surface. Matsui et al (2017), found a stronger oxygen isotope gradient between surface and subsurface foraminifera during the MCO and proposed that the increased gradient resulted from warmer surface waters in the equatorial Pacific relative to subsurface waters. Using typical oxygen isotope change with temperature, the tropical surface waters during the MCO were about 3°C warmer than the subsurface, even if these subsurface waters also warmed because of a deeper thermocline. We expect that surface equatorial waters should have warmed less than those of higher latitudes because of
the ease of heat loss in the tropics through evaporation from warm water to the air.

There is no direct evidence of higher $C_{org}$ degradation in surface waters, but modeling and observations of plankton distribution point to a loss of $C_{org}$ primarily within the surface ocean layers, above 1000 m. Supplemental material from Boscolo-Galazzo et
al. (2021) showed that there were much steeper $d^{13}C$ depth gradients in older time intervals, which modeled to a much shallower and sharper $O_2$ minimum than in the Holocene. Using their temperature dependent model, the Holocene flux of particulate $C_{org}$ at equatorial Pacific Site U1338 was 3 to 4 times greater at 600 m relative to 15 Ma for the same level of primary productivity. Similarly, we observe a factor of 3 to 4 increase in $C_{org}$ sediment content over this same time interval at Site U1338. If this continued through the water column, a much smaller $C_{org}$ flux was sequestered in abyssal waters during the warm MCO climate
relative to modern conditions. Also, it supports a hypothesis that sedimentary $C_{org}$ contents are roughly proportional to the $C_{org}$ particulate rain escaping the surface ocean.

However, $C_{org}$ content of particulate rain that arrives at the abyssal seafloor is significantly larger than the $C_{org}$ buried in the surface sediments, as noted previously. In the modern ocean, the particulate rain captured in deep sediment traps within the




equatorial Pacific region (± 5° of the equator) contains about 5% $C_{org}$ (Table 5 of Honjo et al., 1995) versus the surface sediments that have a $C_{org}$ concentration between 0.23 and 0.33% (Murray and Leinen, 1993). Lower $C_{org}$ MAR away from the equator reflects the lower particulate rain rates away from the equator, and not the lower $C_{org}$ contents in the particulate rain. Differences in $CaCO_3$ rain versus bio-$SiO_2$ rain were observed but did not strongly affect the total $C_{org}$ preserved. In the Holocene, productivity apparently affects the rate of deposition of the particulate rain, but not so much its composition. This also supports the role that the upper water column plays to determine both the magnitude of the biological pump and the level of $C_{org}$ content in sediments.

We note that in periods after the MCO, there is an increase in the ratio of $C_{org}$ to Ba during periods already identified as high productivity intervals. At Site U1338, where we have the best record, high $C_{org}$/Ba reflects periods when productivity was relatively high globally, e.g., the late Miocene Biogenic Bloom (Dickens and Owen; 1999; Diester-Haas et al, 2002; Lyle and Baldauf, 2015; Drury et al 2017; Lyle et al 2019; Karatsolis et al, 2022).

### 5.3.4 Hypothesis (4): A change in proxy relationships for productivity, changing estimates of paleoproductivity

Another hypothesis worth considering is that during the MCO there is a change in the response of the proxy that varies from the expected modern response. For example, diatom deposition or $C_{org}$/Ba ratio might behave differently with respect to $C_{org}$ production. Under these conditions there could be lower actual $C_{org}$ export to the interior ocean than that indicated by the proxy, minimizing the deposition of $C_{org}$ without indicating lower productivity. We believe that there is some likelihood that the relationship between export of particulate $C_{org}$ from the euphotic zone to other biogenic components may be somewhat different under warm earth conditions but propose that these differences result primarily from relative changes in $C_{org}$ consumption in the upper water column.

Dymond and Collier (1996) described how $C_{org}$ rained out of the modern equatorial Pacific relative to Ba. They found lower $C_{org}$ particulate rain away from the equator, corresponded to much lower ratio of $C_{org}$ to Ba (~30). In contrast, this ratio is ~150 near the equator where $C_{org}$ rain was high. The data suggests relatively rapid formation of Ba in microenvironments within particulate rain, followed by a loss of $C_{org}$ versus Ba. There is more complete consumption of $C_{org}$ where the $C_{org}$ flux is lower. The sediment $C_{org}$ to Ba ratios found in sediments (Fig 2) are much lower because of the high degradation of $C_{org}$ at the sea floor relative to Ba (>20x reduction for $C_{org}$ vs ~3x for Ba) before burial. However, the amount of Ba fixed in micro-environments depends on the Ba composition of seawater. In the modern ocean, Ba rain is significantly lower in the Atlantic than in the Pacific relative to $C_{org}$ because of the lower dissolved Ba in the Atlantic (Dymond and Collier, 1996).

If the modern observations across the equatorial region are consistent with changes that might occur in a warm interval like the MCO, we expect lower $C_{org}$ in the particulate rain relative to Ba. Conceivably then, the $C_{org}$ rain might slow even though the Ba flux did not. This could happen if dissolved Ba is fixed into barite ($BaSO_4$) relatively early in the rain of particulates, so that later degradation of $C_{org}$ only affects the $C_{org}$/Ba leaving surface waters (Fig 3). However, the $C_{org}$ MAR resembles the opal MAR to a certain extent after the end of the MCO, indicating that the $C_{org}$ MAR has a profile like a different proxy for productivity when surface waters cool. Perhaps the presence of diatoms causes a more effective transport of particulate $C_{org}$ to the sea floor.

In time series at Site U1338 we find significant non-random changes in the ratio of $C_{org}$ to Ba, associated with apparent changes in productivity (Fig 3). Specifically: high $C_{org}$/Ba associated with the late Miocene Biogenic Bloom, and during a period around



12 Ma that shows high biogenic Si and CaCO₃ deposition as well. These time series show that the biogenic components have their individual processes that lead from creation to burial, so that care needs to be taken to quantitatively ascribe a certain level of primary productivity to the remains found in the sediments. Nevertheless, the lack of $C_{org}$ response during the MCO likely results from upper water column processes.

**6 Conclusions**

In earlier work we have found that warm earth conditions in the Eocene are marked by very low levels of $C_{org}$ burial (Olivarez Lyle and Lyle, 2005, 2006). In this study we also show that warm earth conditions during the Miocene Climate Optimum are also characterized by a low level of $C_{org}$ burial compared to later in the sedimentary record. The low levels are represented in both the weight % $C_{org}$ and $C_{org}$ MAR's, and as low ratios of $C_{org}$ relative to other better preserved biogenic components like

BaSO₄, despite relatively high deposition of other paleoproductivity proxies like biogenic silica. We formed 4 hypotheses to explain the low $C_{org}$ at the MCO: lower productivity, higher degradation in the lower water column, higher degradation in the upper water column, and a change in relationships between proxies, and rejected all except higher degradation the upper water column.

We have identified the upper water column as the region where $C_{org}$ is preferentially removed from the particulate rain to the sea floor, an indication of a 'short circuit' in the biological pump under extreme global warmth once the ocean equilibrates. We observe that the average $C_{org}$ content of equatorial Pacific MCO sediments was about 0.04 % organic carbon, 5 times lower than modern surface sediment of 0.2 to 0.3% $C_{org}$ (Murray and Leinen, 1993). If preserved $C_{org}$ in surface sediments is roughly proportional to the rain of $C_{org}$ that reaches the sea floor, then about 5 times less $C_{org}$ in the particulate rain reached the seafloor at

the MCO, caused by higher metabolic degradation in surface waters.

While such a proportionality of $C_{org}$ in particulate rain is an oversimplification of early diagenesis in pelagic environments, it is an example of how the pelagic sedimentary environment responds to warm earth conditions. Better diagenetic models under low sedimentation rates and oxic conditions might improve our ability to hindcast particulate rain in the past. Clearly, though, there is

a sedimentary response to these processes in the water column.

**Author Contributions**

Mitchell Lyle helped to collect the cores from Sites U1337 and U1338. He organized the XRF analyses along the continuous sediment sections (splices) and found funding for the discrete calibration carbon samples. He had primary responsibility for

writing the paper. Annette Olivarez Lyle trained students, supervised both the bio-SiO₂ and the carbon analyses (CaCO₃ and $C_{org}$) and was responsible for quality control of the results. She also helped write and edit the paper.

**Competing Interests**

The authors declare that they have no conflict of interest.


**Acknowledgements**

We thank the scientific party and crew of the *D/V JOIDES Resolution* on the 2009 Exp 320/321 IODP expedition for collection and initial description of Sites U1337 and U1338. We also thank the students and technical staff that ran the carbon analyses



(Chris Piela, Bianca Romero, Julia Wilson, Anna Stepanova). We also thank Joanna T. Lyle for reviewing the initial draft of this paper.

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
