# Peer review of "Variations in the Biological Pump through the Miocene: Evidence from organic carbon burial in Pacific Ocean sediments."

_Climate of the Past, 2024_

## Author Response (AR1)

Actions taken about referee comments for cp-2024-34 "Variations in the Biological Pump through the Miocene: Evidence from organic carbon burial in Pacific Ocean sediments"

Red marks actions taken in the revision;

blue marks original response to the editors

REF 1 comments

Comments to Lyle and Olivarez Lyle

I have reviewed this manuscript and found very interesting! The new dataset is clearly novel and the discussion is clear and interesting. I have just few remarks and questions and thus ask for minor revisions.

Sincerely

Baptiste Suchéras-Marx

Introduction

l28: Why using Corg rather than POC?

We replaced C-org with POC throughout the manuscript

We shall be happy to change Corg to POC to conform to the preferred abbreviation (POC = Particulate Organic Carbon)

l30-31: You never talk about bioturbation throughout the manuscript. Why so? Could you add argument to exclude this process as consumption of organic matter within the sediment.

We added a new paragraph (paragraph 2) to the Discussion and Implications section basically stating what we state below.

We are unsure what the concern about bioturbation is. All samples are well below the surface sediment mixed layer, where bioturbation primarily occurs. For the most part, bioturbation will smooth the measured parameters of a sedimentary record, reducing the high frequency change. However, it is possible that POC taking longer to pass through the surface mixed layer of 5-10 cm might be exposed to higher degradation, thus reducing the amount of POC that is ultimately sequestered. In the equatorial Pacific there is evidence that more labile POC components are degraded in the upper cm (Stephens et al., 1997, Geochimica et Cosmochimica Acta, Vol. 61, No. 21, pp. 4605-4619), so it is possible that longer residence in the mixed layer might cause higher POC degradation. Because time spent in the mixed layer is proportional to sedimentation rate, we would

expect a correlation between high POC and high sedimentation rates. To examine this possibility, we plotted the POC vs sedimentation rate at Site U1338 and found no correlation (see below). This suggests that bioturbation was not a determining factor in the POC % of these samples.

We can add a paragraph similar to the one above to the manuscript if you deem it important.

[Figure]

Analytical Methods

l120: Could you please add a sentence about uncertainty (2 sigma) for wt%CaCO3 and wt%Corg in order to evaluate the reliability of variations in your record.

We added this into Para 2 in "2. Analytical methods"

This can be added. We have checked consistency by running an in-house sediment standard with each carbon run ("Midway" standard, 2.64±0.02 wt % total carbon, n=523, 0.85±0.01 wt% n=570). We also repeated analyses of every 4th unknown sample during each run day. We monitor both total carbon and organic carbon. The average difference between repeated unknown samples for organic carbon is <0.01 wt%, and if the differences exceeded 0.02 wt % we re-analyzed the sample.

l128: Concerning site 884, the error is up to 0.4wt%. Those data are not shown but comparing to Fig.2, the variations are within this error. Could this bias the use of this site for your interpretations?

We rewrote para 3 in in "2. Analytical methods" to improve clarity.

This sentence was written poorly in our manuscript. We were stating that we used shipboard analyses in addition to our own analyses when CaCO3 was under 15 wt% (1.8% carbonate carbon). The POC contents at Site 884 ranged from 0.1 to 0.4 wt% so the amounts of carbonate carbon and POC were much nearer to each other than if the sediment were 80% CaCO3 (9.6% carbonate carbon) and 0.2 % POC. Shipboard carbon analysis for ODP samples at this time used a coulometer to measure CaCO3 carbon and then subtracted that value from the total carbon measured on a CHN analyser to determine POC. At high CaCO3 values, significant errors often resulted. We added the shipboard analyses because our lab analyses started at about 5 Ma and we wanted to also report on the upper section of the site. The analyses that we did in our lab for Site 884 did

not use the shipboard method, but a much-improved method of measuring organic carbon directly. They have the same analytical precision as reported for Sites U1337 and U1338. The lab samples were also 80% of the reported samples in the interval older than 5 Ma.

We propose to rewrite this section to correct the wrong impression made upon the reviewer. The error in the data from our laboratory is not 0.4wt%.

l149-151: Could you please explain how you estimate the proportion of terrigenous, authigenic oxide and authigenic clay-based Ba at site U1338 in order to evaluate the reliability of the 93% +/- 4% biogenic BaSO4 you have calculated?

We rewrote the last paragraph of "2. Analytical methods " to better explain how we made the estimate.

The estimate is based on normative analyses of clay and ferromanganese oxides from our XRF studies of the sites (see IODP Exp 320/321 Proceedings for more detail on the XRF studies). We assigned a Ba content to each normative component and compared that value to the total Ba measured by XRF. The lowest biogenic Ba occurs where there are the most clays and manganese oxides, which is at the tops of Sites U1337 and U1338.

Age Models and Mass Accumulation Rates (MAR)

l198: You say that "*ages should still be good to +/- 0.2 Ma*". How can you tell? Please describe how you made this estimation.

We added another sentence in section 3.3, stating the sampling-derived uncertainty in the depth/age model.
The age model for Sites 806 and 807 were based on biostratigraphic events, not paleomagnetic datum levels. There are 3 possible errors associated with biostratigraphic datums: mis-identification of species, poor age control on the bio-event, and low resolution in average sample spacing of the biostratigraphic study.

We used bio-events that have proven robust as our age control but had to use the shipboard stratigraphic data and postcruise studies to build an age-depth curve. The sampling density of the data were about 1 sample for every 3 m for foraminifera, and 1 sample for every 4.5 m for calcareous nannofossils. Given sedimentation rates of about 20 m/Myr, the potential error from sampling alone is around 0.14 to 0.22 Myr. We used what we believe is a realistic estimate of total error to be 0.2 Myr.

l203-206: How the sampling density – which is often relatively low resolution for magnetostratigraphy – could influence the reliability of your age model?

We added a description of sample resolution in Section 3.4, where the age/depth model is based entirely on magnetochrons.

The magneto-stratigraphy is based on shipboard measurements using a pass-through magnetometer supplemented by shorebased studies. For Sites U1337 and U1338, see IODP Exp

320/321 Proceedings, where a measurement was made every 2.5 to 5 cm. For Site 884, see ODP Leg 145 Initial Reports and Science Reports, where measurements were made every 10 cm. Magnetostratigraphy on ODP Leg 130 was limited to the Pleistocene.

Results

l220: Your data *"averaged 0.043+/-0.014 wt%"*. This is extremely low. Coming back on previous comment, what are the uncertainties of measurements of Corg?

We added this into Para 2 in "2. Analytical methods"

As previously explained, the $C_{org}$ (POC) measurement has sufficient level of precision and accuracy to make that claim.

Fig. 2: Panel B is in mg/cm2/kyr but in section 3, you said it would be in g/cm2/kyr. Please use the same unit throughout the manuscript. By the way, I would recommend to use g/m2/a, which is more coherent in term of unit (the former Bubnoff unit).

We fixed the label in panel B to g/cm2/ka.

The standard MAR units are either mg or g/cm2/kyr. Using g/m2/yr might make sense for very surficial sediments and comparison to water column fluxes. However, the sedimentation rates are measured cm/kyr or m/Myr, so using the units the reviewer suggested (g/m2/a) would imply much higher precision than we would claim. We prefer to use the standard paleoceanographic units presented in the manuscript. Fig 2 can be changed to g/cm2/kyr if you prefer.

Still in Panel B, DSDP Site 574 show a peak at 12 Ma. Why so? Is it related to a major change in sedimentation rate and thus does the age model is reliable for this interval?

We added a paragraph at the end of section 4 highlighting the changes in POC burial during intervals of very high productivity but not trying to interpret those observations in this paper.

This is a good observation. It is important to note that the same 12 Ma peak can be found in the Site U1338 POC MAR profile as well. Sites U1337 and U1338 have a sedimentation rate peak at about 12 Ma, and since Site 574 is correlated to Site U1337, it does too. Site U1337 was the master site for the U1338 correlation as well because it had a stable isotope record from 20 to 0 Ma. It is very clear where the end of MCO at 13.8 Ma lies based on the stable isotopes, but the carbon and oxygen isotopes are not as distinctive in the 12 Ma interval. We checked the correlation to the CENOGRID isotope stack and did not see any obvious miscorrelation.

Discussion

l373: You site Honjo et al., 1982. With all the respect I have for pioneer work of Honjo, maybe more recent estimation and complex studies have been donw since then.

We added additional references although they didn 't change the point, that particle settling rates are between 50 and 150 m/day.

The settling rate of particles was a minor point to describe how long a typical particle remained in the water column, but we can add newer papers like Berelson (2001), Honda et al (2002), and McDonnell and Buessler (2010). A different settling rate does not change the interpretations we made.

l438: You mentioned that "*modeling and observations of plankton distribution point to a loss of Corg primarily within the surface ocean layers*". This statement about model is circular because models are designed this way, increase temperature = more organic oxidation. Please reevaluate the use of modelling studies in your argumentation.

We went over these statements to insure better clarity.

This is not circular because the studies we referred to were trying to match the past water column stable isotope distribution, and were able to do so only by having temperature dependent oxidation.

REF2 comments

Review of "Variations in the Biological Pump through the Miocene: Evidence from organic carbon burial in Pacific Ocean sediments"

By Lyle and Olivarez-Lyle

Dear Editor,

The manuscript cp-2024-34 by Lyle and Olivarez Lyle follows up their earlier work published in 2006 "Missing organic carbon in Eocene marine sediments: Is metabolism the biological feedback that maintains end-member climates?". In this paper, following the Metabolic Theory of Ecology, the authors for the first time introduced the idea that temperature dependency of metabolic rates may act as a positive feedback to the ocean biological carbon pump on geological time scales. They postulated that under warm climate conditions, such those in the Eocene, enhanced heterotrophs metabolic rates would increase organic matter remineralization in the water column leading to low organic carbon sequestered in deep ocean sediments, explaining lower than expected organic carbon accumulation rates in the equatorial Pacific at this time. This was a consequential paper in paleoceanography, spurring a wealth of studies aimed at investigating the efficiency of the biological pump under different climate states.

With a similar approach, in manuscript cp-2024-34, the authors focus on the trends of biogenic

sedimentary components in the equatorial Pacific from about 21 Ma to present. They present new organic carbon and CaCO3 percentages and accumulation rates from 5 sites and measurements of total Barium for 3 of them. They convincingly show a major shift in the pattern of sedimentation from low biogenic components between 21-14 Ma to higher and variable values from 14 Ma to present. Because the studied sites were always in an approximately equatorial position over the studied interval, a long term decrease in productivity starting at 14 Ma is not a likely explanation. Instead, the authors suggest changes in the efficiency of the biological pump due to the progressive cooling of upper ocean temperatures from the middle Miocene to modern.

I agree with their interpretation and think this is an interesting study which adds to the mounting evidences of a biological pump operating differently depending on background climate, and becoming progressively more efficient with the cooling trend of the last 15 Ma.

I think though, that the message of the paper would be much strengthened by a better contextualization of their data with coeval climatic trends. In particular:

1) The strong link that the authors suggest between the warmth of the Miocene Climate Optimum (17-14 Ma) and low sedimentation of biogenic components is not apparent from the dataset. The dataset shows low sedimentation of biogenic component also for the older interval between 21-17 Ma. Instead, what is really apparent is the increase in biogenic sedimentation from the late middle Miocene on. I would hence recommend the authors not to put so much emphasis on the MCO per se, but rather on the generally warmer early to middle Miocene climate compared to today.

We added western equatorial Pacific data that show that surface waters were warm in the early Miocene as well as the MCO.

We agree that the period prior to the MCO also shows poor preservation of POC and bio-SiO2. We can rewrite the sections that discuss the results to make this point.

2) For the reason above, I also suggest the authors to show their data against climatic records covering the entire age range of their records. For instance, they could additionally show the sea surface temperature record from Auderset et al. 2022 /10.1038/s41586-022-05017-0 from 22 Ma on. This record shows an SST warming trend from about 22 Ma culminating in the MCO, which fits very well with the evidences shown here for low biogenic sedimentation over this whole time interval.

We found an alternate data set that is more pertinent to discussions of the equatorial Pacific, Guillermic et al., 2022. Both BWT and equatorial SST time series are now based on Mg/Ca temperature estimates, so should have the same biases.

Thank you for the Auderset et al. 2022 reference. We have examined the Auderset data and have some concerns. In our review of the data, we found only 3 sites that are relevant to this study: i.e., both within the tropics and during the data period of the MCO. Second, before we would use these data, we would need to check the veracity of the stated paleolatitudes. For example, in

their table Site 806 (Western Equatorial Pacific) remained at its current position (0.3 $^0$N) for 20 million years.  Using the Torsvik et al (2008) Euler rotations for the Pacific Plate, which we assume they have used, would place Site 806 at a paleolatitude of 4.9 °S at 20 Ma, vs 0.3 °N as reported in their table.

3) Using the mid-to low latitude SST record from Auderset et al. 2022, would also help going around the issue of presenting records from an equatorial region against Northern Hemisphere mid latitude temperatures as it is now, which is not ideal and I do not particularly like given other, more suitable records are available.

We replaced the Herbert et al data with Guillermic et al western equatorial Pacific SST.

See the above comment—"low latitude" in the Auderset et al. figure was between 40N and 40S and true low latitude sites were few and far between.

Minor

1) I find confusing that the complete records of data generated are shown only for 2 of the measured sites (U1338 and U1337) (Figs. 3-4). Can you please have the same figure also for the other 3 sites for completeness and to allow an overview on the studied region?

Descriptions and illustrations of time series for Sites 884 and 574 are in the supplemental material, and all data are provided for further study.

We can provide illustrations in the supplemental material for Sites 574, 806, and 807.  Unlike sites U1337 and U1338, we have only POC data and estimates of CaCO3. There is no data for Ba or bio-SiO2.

2) For the same reason I think the data from sites 884 and 1208 should also be shown in Fig. 2, although with the caveat that the sedimentary regime at these sites is less constrained.

We added these data to Figure 2 and revised the figure caption.

We can add Sites 884 and 1208 to Figure 2 but feel that it is more appropriate to show only the equatorial sites.

3) A data availability statement is missing.

We plan to house the data at Pangaea.de after the paper is accepted. We can add this statement at the end of the text.

Typos:

We will fix these typos and any others found.

Line 37 the use of the word "eras" is a bit too colloquial. We changed this to "extended periods".

Boscolo-Galazzo misspelled in lines 38 and 69. Fixed

Boscolo-Galazzo et al. 2018 missing in the reference list. Reference was added

"Total" repeated twice in line 146. Fixed

Reference missing in line 383 for the sentence ending with "abyssal depth". Reference not needed.

"Sediment" repeated twice in line 475. Fixed